# Anaerobiosis and Mutations Can Reduce Susceptibility of *Pseudomonas aeruginosa* to Tobramycin Without Reducing the Cellular Concentration of the Antibiotic

**DOI:** 10.3390/pathogens14020187

**Published:** 2025-02-13

**Authors:** Woravimol Krittaphol, Lois W. Martin, Greg F. Walker, Iain L. Lamont

**Affiliations:** 1School of Pharmacy, University of Otago, Dunedin 9016, New Zealand; pummy.krittaphol@otago.ac.nz (W.K.); greg.walker@otago.ac.nz (G.F.W.); 2Department of Biochemistry, University of Otago, Dunedin 9054, New Zealand

**Keywords:** *Pseudomonas aeruginosa*, tobramycin, resistance, anaerobiosis, *fusA1*, MexXY

## Abstract

Chronic infections of *Pseudomonas aeruginosa* are commonly treated with tobramycin. During infections, the bacteria can exist under conditions of oxygen deprivation that render them less susceptible to this antibiotic. The aims of this research were to investigate the genetic basis of tobramycin resistance under anaerobic conditions, and to investigate the effects of anaerobiosis and mutations on the cellular concentration of tobramycin. Ten mutants with lowered susceptibility to tobramycin than wild-type bacteria were evolved from a laboratory reference strain under anaerobic conditions. Mutations were identified by genome sequencing. Mutations had arisen most frequently in the *fusA1* gene that encodes elongation factor EF-G1A and in genes involved in twitching motility. Cellular concentrations of tobramycin were then measured. Mutations in *fusA1* or absence of the MexXY efflux pump that is associated with tobramycin resistance did not alter the cellular tobramycin concentration under either anaerobic or aerobic conditions. Anaerobic growth reduced the cellular concentration of tobramycin, relative to aerobically grown bacteria, in some but not all of five tested *P. aeruginosa* isolates. Overall, our findings indicate that anaerobiosis and mutations that reduce aminoglycoside effectiveness do not lower the cellular concentration of antibiotic but instead reduce susceptibility through other mechanisms.

## 1. Introduction

The increasing resistance of infectious bacteria to antibiotics poses a major threat to human health and has been described by the World Health Organisation as “a problem so serious that it threatens the achievements of modern medicine”. Occurrence of resistance is particularly problematic in chronic infections, where bacteria are continuously exposed to antibiotics over long periods. One example of such infections is *Pseudomonas aeruginosa*. This bacterium is one of the most frequent and most problematic hospital-acquired pathogens, causing a wide range of acute and hard-to-treat infections [1,2]. It also chronically infects the lungs of individuals with cystic fibrosis (CF) or chronic obstructive pulmonary disease (COPD), impacting on both morbidity and mortality [3]. Antibiotics can suppress the symptoms of infection but once chronic infection is established, antibiotics do not usually eradicate the bacteria.

*P. aeruginosa* can grow under both aerobic and anoxic conditions, using either oxygen or nitrate as an electron acceptor for respiration [4]. During chronic lung infection in individuals with CF or COPD, it grows in biofilms and these, along with host mucus, cause the bacteria to have little or no oxygen [4,5]. Infections are commonly treated with aminoglycoside antibiotics such as tobramycin that inhibit the translation step in protein synthesis [6,7,8]. Under aerobic conditions, susceptibility to aminoglycosides can be reduced by the presence of aminoglycoside-modifying enzymes or by mutations [9,10,11]. Mutations that reduce AG susceptibility have been identified in clinical isolates and also through experimental evolution studies involving in vitro selection of mutants under aerobic growth conditions [12,13,14]. These approaches identified the elongation factor EF-G protein, encoded by the *fusA1* gene, as a key player in resistance, consistent with the detection of *fusA1* mutations in tobramycin-resistant clinical isolates [13,15]. EF-G1A is an essential protein [16] that is required for the translocation step in protein synthesis [17], and substitution mutations in the *fusA1* gene can reduce bacterial growth [11]. However, the molecular mechanisms by which *fusA1* mutations reduce aminoglycoside efficacy have not been determined. Isolates from chronically infected patients also frequently have mutations that result in upregulation of the MexXY efflux pump [18,19,20,21]. Increased expression of this efflux pump is associated with aminoglycoside resistance; however, the relationship between *mexXY* expression and the level of aminoglycoside susceptibility is not clear [22,23]. Although it is generally thought that tobramycin must be a substrate of the efflux pump, direct evidence is lacking.

Experimentally evolved tobramycin-resistant mutants can also have mutations that alter Nuo and Cco proteins that are involved in respiration [12,13], indicating that altered respiration can contribute to resistance. This is consistent with the observation that aminoglycoside efficacy is reduced when *P. aeruginosa* are grown under anaerobic conditions. *P. aeruginosa* isolated from chronically infected cystic fibrosis patients and grown in anaerobic conditions were able to tolerate higher concentrations of tobramycin than the same bacteria grown under aerobic conditions [24]. Other studies have also reported that oxygen deprivation can reduce susceptibility to antibiotics [25,26,27]. Antibiotic efficacy, including that of aminoglycosides, is linked to respiration and the production of toxic reactive oxygen species [28,29], and growth under anaerobic conditions may provide protection against ROS-induced cell death. An alternative explanation for reduced susceptibility in the absence of oxygen is that aminoglycosides enter the cytoplasm of *P. aeruginosa* via a process that is dependent on the membrane potential and is related to respiration [8]. Membrane potential may be reduced under anaerobic conditions or by mutations affecting respiratory proteins, potentially reducing the uptake and, hence, effectiveness of these antibiotics. These lines of evidence led us to hypothesise that altered respiration in the absence of oxygen causes reduced cellular concentrations of tobramycin in *P. aeruginosa*. This study had two main aims. Our first aim was to experimentally evolve *P. aeruginosa* mutants with reduced aminoglycoside susceptibility under anaerobic conditions and compare the mutations that confer resistance with those obtained in aerobically grown bacteria. The second aim of this study was to apply a sensitive assay to accurately measure the cellular concentration of tobramycin to determine if anaerobic growth of *P. aeruginosa* reduces the cellular concentration of tobramycin and to investigate the effects of resistance mutations on cellular tobramycin concentration.

## 2. Materials and Methods

### 2.1. Growth of Bacteria and Antibiotic Sensitivity Testing

The bacteria used in this study are listed in Appendix A. For aerobic growth, bacteria were incubated at 37 °C with aeration (200 rpm) in LB broth, or on LB agar plates. For anaerobic growth, bacteria were grown in LB broth or on LB agar plates containing potassium nitrate (0.4%) in BD GasPak EZ incubation chambers from which the oxygen had been removed using GasPak EZ gas-generating sachets (BD Diagnostics, Sparks Glencoe, MD, USA) following the manufacturer’s instructions. Tobramycin sensitivity testing was carried out using doubling dilutions of antibiotic in Mueller Hinton agar (Becton Dickinson, Auckland, New Zealand) containing potassium nitrate (0.4%) as described previously [24,30], with the minimum inhibitory concentration (MIC) being defined as the lowest concentration of antibiotic that inhibited bacterial growth after 24 h incubation (aerobic growth) or 48 h incubation (anaerobic growth). Tobramycin was purchased from Mylan New Zealand Ltd., Auckland, New Zealand.

### 2.2. Experimental Evolution of Mutants and Identification of Mutations

Mutants with reduced susceptibility to tobramycin were selected using antibiotic gradient plates, as described previously [12], except that the plates were incubated under anaerobic conditions. Briefly, an overnight broth culture of *P. aeruginosa* PAO1 grown from a single colony cultured under aerobic conditions was inoculated onto an LB agar plate containing potassium nitrate (0.4%) and a concentration gradient of tobramycin. Plates were placed in an anaerobic box, the oxygen was removed, and incubation was carried out at 37 °C until colonies were evident (up to 4 days). An isolated colony, growing furthest up the antibiotic gradient, was selected, inoculated into LB broth containing potassium nitrate, and incubated anaerobically at 37 °C overnight. The selection cycle was then repeated using higher concentrations of tobramycin in the antibiotic gradients until mutants that could grow in the presence of higher amounts of tobramycin were not obtained. A single mutant able to grow in the highest amount of tobramycin was purified and analysed further. This procedure was carried out for 10 independent cultures of *P. aeruginosa* PAO1. If two mutants from the same culture had the same tobramycin resistance, one was chosen at random for further analysis.

To identify mutations, whole genome sequencing of mutants was carried out. Genomic DNA was prepared from overnight cultures of end-point mutants using the UltraClean Microbial DNA isolation kit (Mo Bio Laboratories Inc., Carlsbad, CA, USA). Library preparation and sequencing were carried out using the Illumina HiSeq platform to generate 150 bp paired-end reads with at least 100-fold genome coverage by Annoroad Gene Technology Beijing Co., Ltd. (Beijing, China). Mutations were identified by comparison with the parental *P. aeruginosa* PAO1 genome using Breseq [31] as described previously [12,32].

### 2.3. Measurement of Cellular Concentration of Tobramycin

Bacteria were inoculated into LB supplemented with KNO_3_ (0.4%) (25 mL) containing tobramycin (0.4 mg/L) as required. Cultures were incubated overnight at 37 °C under anaerobic conditions and the OD600 was measured. All subsequent steps were carried out at 4 °C. Cells were collected by centrifugation (4000 rpm, 15 min in an Eppendorf 5810R centrifuge). After washing twice in NaCl (0.9%, 1.5 mL), cells were resuspended in 5 mL of NaCl and sonicated on ice (10 bursts of 30 s each with 30 s rests between bursts). This procedure resulted in complete cell lysis, as indicated by clarification of the cell suspensions. The samples were centrifuged (30 min, 4000 rpm) and the supernatants were collected and stored frozen until required. The concentrations of tobramycin were then determined using liquid chromatography tandem mass spectrometry (LC-MS/MS) as described elsewhere [33]. Statistical analysis was carried out using GraphPad Prism5.03 (version 5.03.477) (GraphPad Software, Inc., La Jolla, CA, USA). A two-way ANOVA was performed to compare the means of tobramycin in cell lysates in different bacterial strains and in two different growth conditions. Bonferroni post-tests were then performed to test for interactions between the growth conditions and the type of bacterial strains.

## 3. Results

### 3.1. Characterisation of Mutants with Reduced Susceptibility to Tobramycin Under Anaerobic Conditions

We first applied our antibiotic resistance selection protocol [12] to isolate bacteria with increased resistance to tobramycin when grown under anaerobic conditions. Ten independent mutants were generated that were able to grow in the presence of increased amounts of tobramycin. The MICs of the mutants were determined (Table 1). The mutants had MICs of between two- and sixteen-fold higher than the parental PAO1 strain under anaerobic conditions. They were also less susceptible to tobramycin (two- to eight-fold increase in MIC) under aerobic conditions.

The genomes of the mutants were sequenced and mutations were identified (Table 1). Eight of the ten mutants had mutations in *fusA1*. The *fusA1* gene is also frequently mutated in tobramycin-selected mutants selected under aerobic conditions [12,13,14] as well as in tobramycin-resistant isolates from patients [13,15]. Five of the mutants had mutations in either *pil* or *fimV* genes that are involved in twitching motility. Mutations in other genes were also present in some of the mutants but arose infrequently (one or two mutants), suggesting that they play more minor roles in reducing susceptibility.

### 3.2. Effect of Oxygen Deprivation on Cellular Concentration of Tobramycin

As well as being altered by mutations, susceptibility to tobramycin is reduced during growth under anaerobic conditions (Table 2; [24]).

We have recently developed a novel LC-MS/MS-based assay for measurement of the cellular concentration of tobramycin in *P. aeruginosa* cells [33]. Using this assay, we found that the concentration of tobramycin in cells of the laboratory reference strain PAO1 grown in tobramycin-containing (0.2 mg/L) medium was approximately 10-fold less when cells were grown anaerobically [33]. To determine if anaerobic growth also reduces the cellular tobramycin concentration in clinical isolates of *P. aeruginosa*, the assay was applied to five isolates from patients with CF. The bacteria were grown in the presence of 0.4 mg/L of tobramycin, that is sub-inhibitory for all of the isolates when grown in liquid culture, both aerobically and anaerobically. The amounts of tobramycin in cell extracts were then determined and normalised to sample OD600.

For bacteria grown under aerobic conditions, extracts of cells of the laboratory reference strain PAO1 contained an average of 0.0454 ug/mL tobramycin. This is approximately twice the concentration present in cells grown in the presence of 0.2 mg/L tobramycin (0.0281 mg/L) [33]. A similar amount was present in strain 001C that is phylogenetically closely related to PAO1 [34]. Three of the other isolates contained significantly less tobramycin than strain PAO1, perhaps relating to their higher MICs. However, one isolate, 036-1, had a higher concentration of tobramycin than strain PAO1, showing that increased MIC is not necessarily associated with reduced cellular concentration of the antibiotic.

Under anaerobic conditions, the cellular concentration of tobramycin was reduced by over 10-fold in *P. aeruginosa* strain PAO1, consistent with our previous findings [33]. The cellular concentration of tobramycin in isolates 001C and 036-1 was also significantly lower in anaerobically-grown bacteria. However, there was no significant difference in cellular concentration for strain 015A, and 046-6 had a higher cellular concentration under anaerobic conditions despite having a higher MIC than that under aerobic conditions. The fifth isolate, 006-A2, grew too poorly under anaerobic conditions to allow the assay to be carried out.

### 3.3. Effects of Mutations on Cellular Concentration of Tobramycin

Mutations in the *fusA1* gene increase the MIC of strain PAO1 under aerobic conditions [11,15] and many of the mutants that evolved to have reduced susceptibility to tobramycin under either anaerobic or aerobic conditions have mutations in *fusA1* (Table 1; [12,13,14]). However, the molecular basis of how *fusA1* mutations reduce aminoglycoside susceptibility is not well understood. To test the possibility that *fusA1* mutations alter the cellular concentration of tobramycin, we measured the concentration in three *fusA1* mutants of strain PAO1 (Table 3).

None of the mutants had significantly different amounts of tobramycin from the parental strain during growth under aerobic conditions. In contrast to the parental PAO1 strain, the cellular concentration of tobramycin was not significantly different for any of the three mutants when they were grown under anaerobic conditions, even though the mutants have higher MICs under these conditions. These findings show that the increased MICs of the mutants under anaerobic conditions are not due to reduced cellular concentrations of tobramycin.

Mutations causing upregulation of the MexXYOprM efflux pump are strongly associated with aminoglycoside resistance [19,20,23]. We therefore took advantage of *mexXY* mutants that we have engineered previously [35] to test the hypothesis that this efflux pump expels tobramycin from the bacterial cells. Deletion of *mexXY* did not significantly affect the cellular concentration of tobramycin in reference strain PAO1 or the clinical isolates 015A and 006-A2 under aerobic conditions or, for the first two isolates, under anaerobic conditions. 006-A2 did not grow under anaerobic conditions.

## 4. Discussion

Isolates of *P. aeruginosa* have higher MICs under anaerobic than under aerobic conditions (Table 1; [24,25,26]). The aims of this research were to identify mutations that further reduce tobramycin susceptibility under anaerobic conditions, and to investigate the effects of anaerobiosis and mutations on the cellular concentration of tobramycin. An experimental evolution approach showed that mutations in *fusA1* and mutations affecting twitching motility are associated with reduced susceptibility to tobramycin under anaerobic conditions. Growth under anaerobic conditions had variable effects on the cellular tobramycin concentration, and mutations in *fusA1* did not reduce the cellular concentration, nor did deletion of the *mexXY* efflux pump genes increase it.

Mutations in *fusA1*, which encodes elongation factor EF-G1A, were present in seven of the ten mutants evolved to have reduced tobramycin susceptibility under anaerobic conditions. *fusA1* mutations also arose in bacteria evolved to have reduced susceptibility to tobramycin under aerobic conditions [12,13,32,36,37], and bacteria engineered to have *fusA1* mutations have higher MICs for tobramycin [11,15]. Our findings show that *fusA1* mutations also reduce tobramycin susceptibility under anaerobic conditions. Aminoglycosides bind to ribosomal RNA rather than to EF-G1A [38]; therefore, how *fusA1* mutations reduce aminoglycoside susceptibility is not clear. One possibility, tested here, is that perturbation of cellular processes reduces the cellular concentration of tobramycin. However, our results show that the concentration of tobramycin is not lower in cells of *fusA1* mutants than in those of wild-type bacteria and so the mutations must reduce susceptibility through a different mechanism.

Mutations in *fusA1* can lead to increased expression of the *mexXY* genes [11,39]. The MexXY efflux pump is associated with tobramycin resistance, although direct evidence that tobramycin is a substrate for this efflux pump is lacking. Mutations in *mexXY* did not affect the cellular concentration of tobramycin in three different strains of *P. aeruginosa* under either aerobic or anaerobic conditions (Table 3) even though they increased susceptibility (Table 1 and Table 3; [35]). The lack of effect of *fusA1* mutations on the cellular concentration of tobramycin, even though they are likely to have increased expression of *mexXY*, is consistent with the lack of effect of *mexXY* mutations. These data indicate that tobramycin is not a substrate for the MexXY efflux pump. Instead, it is likely that the effects of changes in MexXY levels on tobramycin susceptibility are indirect, involving alterations to quorum sensing pathways [40]. How changes to EF-G1 reduce susceptibility to tobramycin and other aminoglycosides, beyond their effects on *mexXY* expression, remains unclear although one possibility is that they reduce the affinities of aminoglycosides for their molecular targets in the ribosome [15].

As well as mutations in *fusA1*, half of the evolved mutants had mutations in *pil* and *fimV* genes that are associated with twitching motility. Mutations in twitching motility genes have been identified in a previous study of experimentally evolved tobramycin-resistant mutants [13] and are also associated with reduced susceptibility to ciprofloxacin [12,32,41]. How mutations in *pil* or *fimV* genes could affect aminoglycoside susceptibility under anaerobic conditions is not clear. It also remains to be determined whether *pil* or *fim* mutations are present in clinical isolates of *P. aeruginosa*, and if so, whether they reduce antibiotic effectiveness during infection.

Entry of aminoglycosides across the cytoplasmic membrane and into the cytoplasm is dependent on the membrane potential [8,42,43]. One possible explanation for reduced susceptibility of *P. aeruginosa* to aminoglycosides under anaerobic conditions is that alterations to respiration lower the membrane potential, reducing entry of tobramycin into the bacteria. The presence of mutations altering respiratory enzymes in aerobically evolved resistant mutants [12,36] and the absence of such mutations in the anaerobically evolved mutants (Table 1) are consistent with this explanation. However, although the cellular concentration of tobramycin was significantly higher in three isolates during aerobic growth than in anaerobic growth, this was not the case for one isolate, and for a second isolate, the amount was higher under anaerobic conditions (Table 2). Nonetheless, all isolates had a higher MIC under anaerobic conditions showing that reduced tobramycin susceptibility under these conditions cannot solely be due to lowered cellular concentration of the antibiotic. It seems likely that increased susceptibility under aerobic conditions is due at least in part to the presence of reactive oxygen species that can be generated as a result of antibiotic action and are bactericidal [8,28]. It should also be noted that there were differences between isolates in the cellular concentration of tobramycin under both aerobic and anaerobic growth conditions, although the cellular concentration did not relate directly to the MIC. The effectiveness of other aminoglycosides, on both *P. aeruginosa* and other species, is also reduced under anaerobic conditions [27] and it will be of importance to determine the cellular concentrations of other antibiotics, in other species.

In conclusion, this work shows that the increased MIC of tobramycin under anaerobic conditions is not necessarily due to lower cellular amounts of tobramycin. Instead, it is consistent with antibiotic efficacy being due in part to the generation of reactive oxygen species following antibiotic exposure. *fusA1* mutations are a prime contributor to reduced tobramycin susceptibility under anaerobic conditions, but mutations in *fusA1* or *mexXY* do not, in general, significantly change the cellular concentration of tobramycin, indicating that these mutations reduce tobramycin susceptibility through a different mechanism. It will be of interest to determine the cellular concentrations of other antibiotics, including in other bacteria, under both aerobic and anaerobic conditions.

## Figures and Tables

**Table 1 pathogens-14-00187-t001:** MICs and mutations for experimentally evolved mutants.

Mutant	MIC *		Mutations		
	aerobic	anaerobic	*fusA1*	*pil*/*fimV*	Other
PAO1 (wild-type)	0.5	2			
TanA3	1	4	D373Y		
TanB3	1	4			
TanC3	1	8	M431K	*fimV* E449 *	*PA3943* Y60C
TanD3	4	8	T456A	*pilZ* 2 bp deletion	
TanE3	1	6	Y690C		*rpmL* 1 bp deletion
TanF3	1	4		*fimV* Q234 *	*pmrB* R287Q
TanG5	2	16	E100G		*dnaX* 6 bp insertion *mexY* V398A
TanH4	2	16	V548A	*pilB* T333M	
TanI5	2	32	V488A		*mgtE* M438I *foxA* S563G *pssA* D240G *PA5121* I516V
TanJ3	1	4		*fimV* 1 bp insertion	*pmrB* R287Q

* MIC values are medians of at least 3 biological replicates. All biological replicate values are shown in Appendix A.

**Table 2 pathogens-14-00187-t002:** MIC values and concentrations of tobramycin in *P. aeruginosa*.

Isolate	MIC *		Tobramycin Concentration (mg/L) ^†^		Aerobic v Anaerobic *p* Value	Difference from PAO1 (*p* Value)	
	aerobic	anaerobic	aerobic	anaerobic		aerobic	anaerobic
PAO1	0.5	2	0.0454 ± 0.0106	0.0041 ± 0.0018	<0.001	na	na
001C	4	8	0.0459 ± 0.0123	0.0103 ± 0.0031	<0.01	ns	ns
015A	4	16	0.0151 ± 0.0017	0.0166 ± 0.0148	ns	<0.01	ns
036-1	1.5	32	0.0679 ± 0.0102	0.0296 ± 0.0044	<0.001	<0.05	<0.05
046-6	8	24	0.0224 ± 0.0059	0.0810 ± 0.0183	<0.001	<0.05	<0.001
006-A2	16	64	0.0100 ± 0.0029	nd	na	<0.001	na

* The MICs of isolates 015A and 006-A2 were determined in this study, with biological replicate values shown in Appendix A. Those of the remaining isolates were reported previously [24]. ^†^ Amount of tobramycin in cell lysates, normalised to culture OD600. Values are mean ± SD of 3 biological replicates. Values of individual replicates are shown in Appendix A. na, not applicable; ns, not significant; nd, not determined as the bacteria did not grow under these conditions.

**Table 3 pathogens-14-00187-t003:** MIC values and concentrations of tobramycin in *P. aeruginosa* mutants.

Isolate	MIC		Tobramycin Concentration (mg/L) *		Aerobic v Anaerobic (*p* Value)	Difference from Wild-Type (*p* Value)	
	aerobic	anaerobic	aerobic	anaerobic		aerobic	anaerobic
PAO1	0.5	2	0.0454 ± 0.0106	0.0041 ± 0.0018	<0.001	na	na
PAO1 *fusA1* R680C	2	8	0.0268 ± 0.0180	0.0144 ± 0.0079	ns	ns	ns
PAO1 *fusA1* D327S	4	16	0.0472 ± 0.0111	0.0513 ± 0.0335	ns	ns	<0.001
PAO1 *fusA1* Y683D	2	8	0.0374 ± 0.0188	0.0245 ± 0.0041	ns	ns	<0.05
PAO1 Δ*mexXY*	0.25	2	0.0454 ± 0.0083	0.0112 ± 0.0022	<0.01	ns	ns
015A Δ*mexXY*	1	4	0.0149 ± 0.0009	0.0122 ± 0.0066	ns	ns	ns
006-A2 Δ*mexXY*	1	8	0.0157 ± 0.0028	nd	na	ns	na

* Amount of tobramycin in cell lysates, normalised to culture OD600. Values are mean ± SD of 3 biological replicates. Values of individual replicates are shown in Appendix A. na, not applicable; ns, not significant; nd, not determined as the bacteria did not grow under these conditions.

## Data Availability

All data are contained within this article and Appendix A.

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
