# Peer review of "Anaerobiosis and Mutations Can Reduce Susceptibility of Pseudomonas aeruginosa to Tobramycin Without Reducing the Cellular Concentration of the Antibiotic"

_pathogens, 2025, doi:10.3390/pathogens14020187_

Round 1

Reviewer 1 Report

Comments and Suggestions for Authors

The authors detected several mutations with higher tolerance to tobramycin and assessed the intracellular concentration of tobramycin under both aerobic and anaerobic conditions for five clinical isolates. They found that a higher MIC does not necessarily indicate a reduced intracellular tobramycin concentration, and PA appears to have a higher MIC under anaerobic conditions compared to aerobic conditions. The results support the conclusions, and the MS data seem very reliable. However, I have some suggestions and questions:

  1. In vivo and in vitro should be italicized.

  2. In Table 1, why did the authors choose the Tan A B C D E F G H I J mutants as the evolved mutants? An explanation should be provided in the manuscript or in the legend.

  3. In Tables 2 and 3, why did the authors not check the concentration of the mutations they identified through their own experiments, but instead used clinical isolates?

Author Response

The authors detected several mutations with higher tolerance to tobramycin and assessed the intracellular concentration of tobramycin under both aerobic and anaerobic conditions for five clinical isolates. They found that a higher MIC does not necessarily indicate a reduced intracellular tobramycin concentration, and PA appears to have a higher MIC under anaerobic conditions compared to aerobic conditions. The results support the conclusions, and the MS data seem very reliable. However, I have some suggestions and questions:

-> Thank you to the reviewer for their careful and constructive assessment of our manuscript. We consider that addressing these points has improved the manuscript.

  1. In vivo and in vitro should be italicized.

-> Thank you for pointing this out. We have now italicised in the text and also the references. 

2. In Table 1, why did the authors choose the Tan A B C D E F G H I J mutants as the evolved mutants? An explanation should be provided in the manuscript or in the legend.

->  The mutants were chosen as having arisen independently, ie. in different cultures, to avoid multiple isolations of the same mutant. The data in Table 1 showing the mutations present in each mutant make it clear that the mutants are not related to each other ie. are indeed independent. One mutant was selected from each of 10 cultures. For each culture, the mutant that was chosen was one that showed maximum resistance on the antibiotic gradient plate (ie. grew furthest up the gradient) as described in the Methods (lines 109-115). We have expanded the methods section (lines 110-115 in the revised manuscript) to explain this procedure in more detail and address the reviewer’s point.

3. In Tables 2 and 3, why did the authors not check the concentration of the mutations they identified through their own experiments, but instead used clinical isolates?

->  The purpose of the experiments described in Table 2 was to determine the effects of anaerobiosis on the intracellular concentration of tobramycin in isolates of P. aeruginosa. All of the mutants described in Table 1 were from the widely used reference strain PAO1. However we considered it important to determine whether strain PAO1 is representative of all P. aeruginosa isolates. The data in Table 2 indicate that some but not all strains behave in the same way as PAO1, which is an important finding when considering the data in relation to infections with P. aeruginosa.
The purpose of the experiments described in Table 3 was to determine the effects of mutations in genes that are strongly associated with tobramycin resistance (fusA1 and mexXY) on the intracellular concentration of antibiotic.  The mutants used are identical to the parental wild-type strains except for the presence of the defined fusA1/ mexXY mutations, meaning that any differences between mutant and wild-type strains must be due to the mutations. We considered the option of analysing the mutants that were isolated in this study and are described in Table 1. However as most of the mutants have multiple mutations, it would not be possible to ascribe differences to the effects of specific mutations.

Reviewer 2 Report

Comments and Suggestions for Authors

This study aims to investigate the selective pressure of anaerobiosis on naturally occurring genetic mutations in Pseudomonas aeruginosa that modulate bacterial susceptibility to aminoglycoside antibiotic tobramycin. The antibiotic tolerance of opportunistic pathogens presents serious health problem and demands extensive research that can help with the design of more efficient therapeutics.

Although lacking innovation and proper methodology in some cases, this work has a potential to contribute to the collective knowledge on the mechanisms of bacterial tolerance to antibiotics. This contribution itself warrants publication should the authors be willing to put some additional work and provide new evidence on the mechanisms of FusA1 protein mutations and their effect on Pa susceptibility to aminoglycoside antibiotics.  

The authors have stated two main aims of this study: 1) Experimentally evolve P. aeruginosa mutants with reduced aminoglycoside susceptibility under anaerobic conditions; and 2) Determine if anaerobic growth of P. aeruginosa reduces the cellular concentration of tobramycin, and to investigate the effects of resistance mutations on the cellular tobramycin concentration.  

Regarding aim 1, selection of antibiotic tolerant bacterial species under specific pressure (in this study, anaerobic conditions and antibiotic concentration gradient) cannot be considered an evolution since one cannot proof with the methods applied in this study that the mutant bacterial strains were not present in the population before the selective pressure was applied. One way to provide such proof is to start the passages from a clonal bacterial population that has been sequenced before the selection pressure and after.

The results presented here confirm previously known fact that EF-G protein, encoded by the fusA1 gene, is a key player in resistance to tobramycin since majority of the sequenced clones have mutations in the fusA1 gene. Finding mutations at similar rate in genes encoding motility is the actual novelty of the study that complements previous knowledge linking bacterial motility to the dynamics of antibiotic resistance evolution (Nat Commun. 2023 Sep 11;14:5584. doi: 10.1038/s41467-023-41196-8 ). Interestingly, double mutants for fusA and fimV/fil have double the antibiotic tolerance at anaerobic conditions.

However, this study lacks information on the effect of mutations on motility. Do the mutations increase or decrease motility? Similarly, no effort was made to determine whether the mutations in the fusA1 gene affect the function of the EF-G protein.

To increase the impact of the study, two additional experiments are suggested:

1)     Motility assay for TanF3 or TanJ3 compared to wt PAO1. This will require a simple experiment in low density agar and antibiotic concentration gradient. Bacteria can be fluorescently labeled with FITC to increase sensitivity of single bacterial cells detection. This experiment can be done in aerobic and anaerobic conditions.

2)     Effect of fusA1 mutations on total protein content can be easily measured with a colorimetric assay of bacterial culture lysate and comparison can be made between mutant and wet cultures at aerobic and anaerobic conditions. Additionally ROS accumulation in mutants and wt cultures when tobramycin is present can be measures with a fluorescent assay.

Regarding aim2, the claim, that highly sensitive assay was applied to determine if anaerobic growth of P. aeruginosa reduces the cellular concentration of tobramycin, is an overstatement. Yes, the LC-MS/MS is a sensitive method however the quantitative analysis in this study is not reliable based on the normalization of intracellular antibiotic concentration to OD600. Ultrasonication will not efficiently break bacterial cells, especially stressed by oxygen deprivation that thickens cell wall. Canadian press will do the job or cell lysis. Normalization of the amount of released antibiotic must be done to an intracellular metabolite (ATP will be the easiest choice).

Author Response

Thank you to the reviewer for their careful and considered assessment of our manuscript. With regard to the specific points made by the reviewer:

  1. Regarding aim 1, selection of antibiotic tolerant bacterial species under specific pressure (in this study, anaerobic conditions and antibiotic concentration gradient) cannot be considered an evolution since one cannot proof with the methods applied in this study that the mutant bacterial strains were not present in the population before the selective pressure was applied. One way to provide such proof is to start the passages from a clonal bacterial population that has been sequenced before the selection pressure and after.

->  The term “experimental evolution”, or similar terms, is commonly applied to the  approach of exposing bacteria to an environmental stress, such as antibiotic exposure, and selecting for mutants better able to tolerate the stress (eg. references 12-14 in the manuscript). The ten mutants analysed in this study were each selected from an independent culture established from a founder strain that had been sequenced previously, as described in the manuscript. We completely agree that it is possible that mutants were present in the founder populations, although as different mutants were obtained from each culture, it seems more likely that each arose independently during growth of the culture. The purpose of the experiment was to identify mutations that can reduce susceptibility to tobramycin under anaerobic conditions, and from that perspective it does not really matter whether the mutants were present at the start of the culture or arose during selection.

2.   To increase the impact of the study, two additional experiments are suggested:
           (1) Motility assay for TanF3 or TanJ3 compared to wt PAO1. This will require a simple experiment in low density agar and antibiotic concentration gradient. Bacteria can be fluorescently labeled with FITC to increase sensitivity of single bacterial cells detection. This experiment can be done in aerobic and anaerobic conditions.

-> Thank you for the suggestion. We fully agree that it will be important to determine the effects of fim and pil mutations on both motility and antibiotic resistance. However the mutants developed in this study also contain mutations in other genes (fusA1, pmrB etc). The presence of these mutations would mean that differences in motility from wild-type could not be attributed solely to the fim/pil mutations. It will be necessary to engineer mutants that are identical to the wild-type PAO1 strain except for the fim/pil mutations, to clearly identify the effects of the mutations. This work, along with a full characterisation of the effects of the mutations, is beyond the scope of the present study which is a Brief Report restricted to addressing the aims that are nicely summarised by the reviewer.

3.  (2)  Effect of fusA1 mutations on total protein content can be easily measured with a colorimetric assay of bacterial culture lysate and comparison can be made between mutant and wet cultures at aerobic and anaerobic conditions. Additionally ROS accumulation in mutants and wt cultures when tobramycin is present can be measures with a fluorescent assay.

->  We appreciate the suggestion of measuring protein content as an approach to investigating the effects of fusA1 mutations. This approach would give insights into the effects of these mutations on the amount of cellular protein and will be worth keeping in  mind for investigations into the global effects of fusA1 mutations. However this Brief Report if focussed on tobramycin resistance, including addressing the question, is fusA1 associated with resistance under anaerobic conditions? The more global effects of fusA1 mutations are beyond the scope of the study. 
An implication of our study is that the effects of tobramycin under aerobic conditions are due in part to the presence of ROS species under these conditions, but not under anaerobic conditions. Whether tobramycin increases the amount of ROS under aerobic conditions is an interesting question, but it is beyond the scope and aims of this study. 

4.  Regarding aim2, the claim, that highly sensitive assay was applied to determine if anaerobic growth of P. aeruginosa reduces the cellular concentration of tobramycin, is an overstatement. Yes, the LC-MS/MS is a sensitive method however the quantitative analysis in this study is not reliable based on the normalization of intracellular antibiotic concentration to OD600. Ultrasonication will not efficiently break bacterial cells, especially stressed by oxygen deprivation that thickens cell wall. Canadian press will do the job or cell lysis. Normalization of the amount of released antibiotic must be done to an intracellular metabolite (ATP will be the easiest choice).

->  Thank you for the suggestion. The use of an intracellular metabolite as an internal standard for measurement of tobramycin concentration is in principle an attractive approach. A difficulty is that concentrations of metabolites are affected by growth conditions and could also conceivably be affected by mutations associated with antibiotic resistance. For example, the cellular concentration of ATP is significantly reduced during anaerobic growth (Armitage and Evans, FEBS Lett 156:113-118 [1983]; Tran and Unden, Eur J Biochem 251:538-543 [1998]) which would confound the analysis if this were used as an internal standard. We have used OD600 for normalisation in our procedure as it provides an overall view of the amount of bacterial biomass that is present. In our hands, the sonication procedure gives a reproducible amount of cell lysis and clarified cell suspensions, indicating that lysis was essentially complete. We have now added this information to the manuscript (line 130-131). Independent biological replicates of the same isolate/ mutant (Table S3) show good reproducibility indicating that our sonication lysis technique gives reproducible results.

Round 2

Reviewer 2 Report

Comments and Suggestions for Authors

I am recommending this work for publication due to the diligence of the Corresponding Author/s to engage in discussion and address some of my concerns

Should the authors have addressed some of the suggestions to show clear link between mutations and function, the work would have been more impactful. It may be the case that the research team do not have the funding to perform additional experiments.

Since the research presented here is not wrong and the discussion of the results is mostly correct (although in some cases presumptive) I would not impede reporting of this work to the scientific community.